# Proposal

**Kehan Zheng**
Tsinghua University

**Yida Lu**
Tsinghua University

**Wenjing Wu**
Tsinghua University

## 1 Background

College students often face challenges in getting enough personalized practice, which hinders their ability to thoroughly grasp complex material and improve their academic performance. This issue is particularly pronounced in courses like physics and engineering, where students are required to develop a deep conceptual understanding alongside strong problem-solving skills. Traditional methods of teaching and assessment often fall short in providing the individual attention that students need, especially in large classes where personalized feedback is limited.

As education technology rapidly advances, there is an increasing need to explore how cutting-edge tools, such as artificial intelligence, can transform traditional learning methods. AI has the potential to provide tailored learning experiences at scale, offering solutions to long-standing issues in education. The goal of our project is to address this gap by developing an AI-powered teaching assistant based on ChatGLM, which will offer students personalized practice and learning resources.

The AI teaching assistant we envision will be capable of generating customized questions and exercises aligned with specific topics or areas where a student may need additional practice. Unlike generic practice platforms, our system will adapt to the individual needs of each user, creating a dynamic and interactive learning experience. To begin, our focus will be on university-level courses such as Physics and Chemical Engineering Thermodynamics—subjects that require both theoretical knowledge and the ability to apply concepts in problem-solving scenarios.

This project exemplifies the practical application of AI in addressing real-world educational challenges. By personalizing learning through a large language model, we aim to bridge the gap between classroom instruction and individual student needs. If successful, this approach could have far-reaching implications, revolutionizing the way students interact with educational content. Personalized AI-driven practice could empower students to work at their own pace, receive instant feedback, and strengthen their understanding of difficult topics.

## 2 Definition

Setting Questions is a text generation task where, given specific requirements such as knowledge points, testing formats, and question types, a model is designed to produce questions that meet these criteria and can be answered effectively. However, current models often face significant issues, such as missing variables and poor association with the intended knowledge points. This project aims to improve the model's question-setting performance through various optimization techniques, enhancing its practical applicability.

## 3 Related Work

**LLMs for Text Generation**  Text Generation aims at producing coherent natural language responses to human input and is a fundamental task for language models [9]. With the advancement of machine learning techniques, large language models (LLMs) have achieved remarkable performance across multiple text generation tasks [2, 14, 3]. LLMs, such as GPT-4 [1], typically adopt a decoder-only architecture and employ next token prediction objective, which is well-suited for text generation.

Preprint. Under review.

In consequence, these models maintain an outstanding capability on generation tasks. In contrast, GLM [4] unifies different pre-training architectures with autoregressive blank filling, demonstrating comparable performance to decoder-only models in text generation. To facilitate the application and development of LLMs, many open-sourced LLMs have emerged [12, 7] and are widely used for academic or commercial purpose. Our work leverages LLMs' impressive text generation ability and utilizes open-sourced LLMs as AI teaching assistants.

**Methods for Specific Task Generation**   As LLMs are trained on general objectives, it is essential to align LLMs with specific downstream tasks. A common method involves prompting LLMs for task completion [5, 10]. However, improper prompts may result in suboptimal performance [8]. To better adapt LLMs to the downstream tasks, supervised fine-tuning (SFT) is often employed [11, 13]. SFT leverages labeled data from a given task and trains an LLM with Cross Entropy loss. While SFT is effective in enhancing LLMs' task-specific performance, it demands a significant amount of high-quality data, which can be difficult to obtain in certain fields. Another approach is Retrieval-Augmented Generation (RAG), which assists LLMs with retrieved passages [6]. RAG retrieves relevant documents according to the input and provides the results to LLMs, thereby facilitating more accurate responses. RAG improves the reliability of the generated results and reduces training costs. In this work, we primarily utilize RAG and SFT to align LLMs to teaching assistant tasks.

# 4   Proposed Method

We have selected the GLM-4 series models for our project, which are known for their strong foundational capabilities and good instruction-following abilities. Currently, we have datasets of questions from two subjects: University Physics and Chemical Thermodynamics, each containing over a thousand entries. The richness of these datasets provides a sufficient basis for carrying out this task.

For the evaluation framework, we aim to establish a comprehensive set of criteria to assess question quality. The evaluation will focus on aspects such as knowledge point matching, solvability, completeness of the question setting, and difficulty level. We intend to develop an automated assessment approach to quantify the quality of generated questions based on these standards. Using this evaluation system, we have chosen the open-source GLM-4-9B-Chat model as our baseline.

In terms of methods, we plan to experiment with two primary approaches: Retrieval-Augmented Generation (RAG) and Supervised Fine-Tuning (SFT). For RAG, we aim to implement a related module that retrieves relevant questions based on the given requirements, incorporating them into the prompt to guide the model in mimicking or rewriting. In SFT, our focus is on fine-tuning the GLM-4-9B-Chat model to enable a direct conversion from "question requirements" to "questions." Both methods have demonstrated effectiveness in text generation tasks, which is why we chose them. Additionally, we will consider incorporating more advanced techniques in the future, such as providing explicit reasoning chains or adding suggestions for modifications if they prove to enhance performance.

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
