# OpenReview forum: "【Proposal】"
_tsinghua.edu.cn/THU/2024/Fall/AML — THU 2024 Fall AML Submission_

### Official Review · ~Bryan_Constantine_Sadihin1 · 2024-11-09
**Review of "Proposal"**

**Rating:** 9
**Confidence:** 4

**Review:**

Strength:
1. Educational Relevance: This proposal innovatively uses AI-powered teaching assistant to targets challenging subjects such as physics and engineering. The success of this project is highly relevant to modern education.
2. Comprehensive Methodological Approach and Evaluation Framework: The proposal outlines a detailed strategy to solve the research problem and evaluation framework.

Weakness:
1. Paper formatting: Some format of the papers can be improved, such as nonexistent research title and abstract.

---

### Official Review · ~Keyu_Shen1 · 2024-11-10
**Good Proposal**

**Rating:** 8
**Confidence:** 3

**Review:**

The proposal aims to develop a system that generates customized questions tailored to individual learning needs, leveraging techniques such as Retrieval-Augmented Generation (RAG) and Supervised Fine-Tuning (SFT). Strengths of the proposal include its clear relevance to current educational challenges, and the integration of RAG and SFT demonstrates the use of advanced techniques to align the model with educational tasks. However, the proposal could benefit from a more explicit discussion on potential challenges, such as data insufficiency, the quality of generated questions, and the design of evaluation metrics. Additionally, the proposal could establish its novelty more clearly to distinguish its contributions within the field.

---

### Official Review · ~Junjie_Chen1 · 2024-11-11
**Good Work**

**Rating:** 8
**Confidence:** 4

**Review:**

The proposal effectively addresses a crucial challenge in higher education by proposing an AI-powered teaching assistant to provide personalized practice for students. Its structure is clear, and the focus on using advanced models like GLM-4 demonstrates feasibility. The inclusion of methods such as Retrieval-Augmented Generation (RAG) and Supervised Fine-Tuning (SFT) aligns well with the stated goals, and the proposal is grounded in practical applications with rich datasets from university-level courses.

However, the baselines provided, such as the use of GLM-4-9B-Chat, seem limited for comparison and lack diversity in competing approaches. Additional baselines incorporating alternative models or methodologies would strengthen the evaluation.

---

### Official Review · ~Nan_Sun10 · 2024-11-11
**Ambitious but Unrefined: An AI-Powered Teaching Assistant Proposal for Personalized Student Practice**

**Rating:** 8
**Confidence:** 3

**Review:**

This proposal introduces an AI-powered teaching assistant based on the GLM-4 series models, aimed at generating personalized practice questions for university students in physics and chemical engineering thermodynamics.

The idea holds significant promise, particularly in addressing the lack of individualized practice in large classroom settings, which often leads to gaps in student understanding. By leveraging techniques like Retrieval-Augmented Generation and Supervised Fine-Tuning, the system aspires to align generated questions with specific knowledge points and student needs.

However, the proposal lacks a detailed plan for scaling the solution across diverse academic subjects and adapting to individual student progress over time. The reliance on RAG and SFT, while effective, might not fully address the intricacies of maintaining question quality and consistency without further enhancements. Additionally, the evaluation metrics are not fully developed, which could lead to challenges in objectively assessing the effectiveness of generated content.

---

### Official Review · ~Tianhai_Liang1 · 2024-11-11
**Good Proposal**

**Rating:** 8
**Confidence:** 4

**Review:**

The proposal aims to develop an AI-powered teaching assistant using ChatGLM to provide personalized practice for university courses like Physics. It uses Retrieval-Augmented Generation (RAG) and Supervised Fine-Tuning (SFT) to improve question generation and includes automated evaluation for quality control.

The method supports personalized learning with adaptive questions, improving student engagement and learning efficiency. It leverages advanced AI techniques to optimize education, making practice more interactive and scalable.

The approach requires large labeled datasets for SFT, which can be hard to obtain. There’s also a risk of generating off-target content, and it demands substantial technical resources, raising costs and complexity. Also, this proposal would be better if there was a clear title and abstract.

---

### Official Review · ~Yang_Ouyang2 · 2024-11-11
**Great background research and methodology, but does not address scaling and privacy concerns.**

**Rating:** 9
**Confidence:** 5

**Review:**

Strengths
Clear  Motivation and objectives: The projects has a clear purpose and motivation.
Solid Background: The background section and related work are well-researched.
Detailed Methodology and Evaluation Plan.

Weaknesses
Personalization Mechanism: Need more details on the personalization mechanism
Data Privacy: how and what data will be collected?

The proposal shows a promising solution for personalized learning, but could improve by addressing the personalization mechanism, data privacy, and scalability.

---

### Official Review · ~Kaiwei_Zhang3 · 2024-11-12
**Has pratical value, yet lacks novelty and difficulty**

**Rating:** 6
**Confidence:** 5

**Review:**

**1. Summary:**

The proposal aims to develop an AI-powered teaching assistant using the GLM-4 model to provide personalized practice questions for university-level courses, initially focusing on Physics and Chemical Engineering Thermodynamics. It highlights the need for individualized practice in challenging subjects, where traditional methods lack tailored feedback for large student groups.

**2. Clarity:**

The paper is fairly clear on its purpose. However, it would be better to furthur illustrate on how to improve the performance of AI teaching assistants.

**3. Originality:**

The idea of training an AI assistant for classes has long been proposed, and various products have been published.

**4. Significance:**

Using AI to assist education and teaching is a very hot subject nowadays, and is of great significance since it could leverage the inequality of educational resources, improve studying efficiency, and reduce the burden of teachers.

**5. Pros:**
* The project focuses on a practical problem, and could potentially be used to aid the studying of students in Chemical Thermodynamics and University Physics classes.

**6. Cons:**
* There are already a huge amount of AI teaching assistants. It would be better to summarize current developments in this field and analyze the weakness of existing products.
* The project is fairly easy. Fintuning LLMs or doing RAGs is not considered a complex task.
* The authors point out that "current models often face significant issues, such as missing variables and poor association with the intended knowledge points." However, they do not explain how to solve these existing problems. Furthur investigations about SFT and RAG techniques are needed.

---

### Official Review · ~Eddy_Yue1 · 2024-11-12
**Chem & Phys**

**Rating:** 8
**Confidence:** 4

**Review:**

Due to the large amounts of AI teaching assistants on the rise, the idea of specialising in Uni chem and phys and leveraging ChatGLM to provide targeted practice, along with the focus on practice question generation may give this bot an advantage.

Writing a more elaborate and specific title could definitely go a long way.

---

### Official Review · ~Maanping_Shao1 · 2024-11-12

**Rating:** 8
**Confidence:** 3

**Review:**

This proposal, presented by researchers from Tsinghua University, introduces an AI-powered teaching assistant leveraging the ChatGLM model to provide personalized practice for university students in challenging subjects like physics and engineering thermodynamics. The assistant generates tailored exercises, addressing gaps in traditional education by offering individualized learning experiences. The proposed methods include Retrieval-Augmented Generation (RAG) and Supervised Fine-Tuning (SFT) to improve alignment with specific educational needs. This approach has the potential to significantly enhance students' understanding by integrating AI-driven, adaptable support into their learning process.

---

### Official Review · ~Kuanghao_Wang1 · 2024-11-12
**Good direction**

**Rating:** 8
**Confidence:** 4

**Review:**

The research for this proposal is on AI teaching assistants, which argues that science and engineering undergraduates face challenges in getting enough personalised practice, which prevents them from deepening their understanding and improving their academic performance, and that AI teaching assistants can be an effective solution to this problem. This proposal will be based on Chat-GLM, using RAG and SFT methods to improve the task-specific performance and generalisation of the model. The paper describes the problem more clearly and gives more specific follow-up steps. However, it should be noted that there are already many AI assistants on the market, and this paper should highlight what advantages and shortcomings the results of this paper will have compared to the existing ones. In addition, the format of the article is slightly problematic, and needs to be supplemented with a clearer title.